# ECMamba: Consolidating Selective State Space Model with Retinex Guidance for Efficient Multiple Exposure Correction

**Wei Dong**[1,*]**, Han Zhou**[1,*]**, Yulun Zhang**[2]**, Xiaohong Liu**[2,†]**, Jun Chen**[1]

[1]McMaster University    [2]Shanghai Jiao Tong University

{dongw22, zhouh115, chenjun}@mcmaster.ca    yulun100@gmail.com    xiaohongliu@sjtu.edu.cn

[*]Equal Contribution    [†]Corresponding Author

## Abstract

Exposure Correction (EC) aims to recover proper exposure conditions for images captured under over-exposure or under-exposure scenarios. While existing deep learning models have shown promising results, few have fully embedded Retinex theory into their architecture, highlighting a gap in current methodologies. Additionally, the balance between high performance and efficiency remains an under-explored problem for exposure correction task. Inspired by Mamba which demonstrates powerful and highly efficient sequence modeling, we introduce a novel framework based on **Mamba** for **E**xposure **C**orrection (**ECMamba**) with dual pathways, each dedicated to the restoration of reflectance and illumination map, respectively. Specifically, we firstly derive the Retinex theory and we train a Retinex estimator capable of mapping inputs into two intermediary spaces, each approximating the target reflectance and illumination map, respectively. This setup facilitates the refined restoration process of the subsequent **E**xposure **C**orrection **M**amba **M**odule (**ECMM**). Moreover, we develop a novel **2D S**elective **S**tate-space layer guided by **Retinex** information (**Retinex-SS2D**) as the core operator of **ECMM**. This architecture incorporates an innovative 2D scanning strategy based on deformable feature aggregation, thereby enhancing both efficiency and effectiveness. Extensive experiment results and comprehensive ablation studies demonstrate the outstanding performance and the importance of each component of our proposed ECMamba. Code is available at `https://github.com/LowlevelAI/ECMamba`.

## 1 Introduction

Images captured under over-exposure and under-exposure conditions suffer from various degradations, including reduced contrast, color distortion, and information loss in extremely dark or bright regions. The objective of exposure correction is to enhance the visibility, contrast, and structural details for images with various illumination conditions, which is pivotal for improving the performance of a plethora of downstream applications such as object detection, tracking, and segmentation systems [10, 39, 31] in scenarios with improper exposure.

Similar to other image restoration tasks [3, 2, 32, 25, 26, 23, 42, 12], many deep learning models [38, 6, 34, 44] have been proposed for under-exposed image enhancement and demonstrate commendable results. However, our preliminary experiments indicate that these methods generally perform poorly in multi-exposure correction. This inadequacy stems from the distinct mapping flow between Over-Exposed (OE) and Normal-Exposed (NE) images compared to that between Under-Exposed (UE) and NE images. Recently, some promising works [20, 21, 1, 19] have introduced several interesting deep learning networks to learn consistent exposure representations for multi-exposure correction. However, despite its widespread adoption and outstanding performance in under-exposure correction,

38th Conference on Neural Information Processing Systems (NeurIPS 2024).

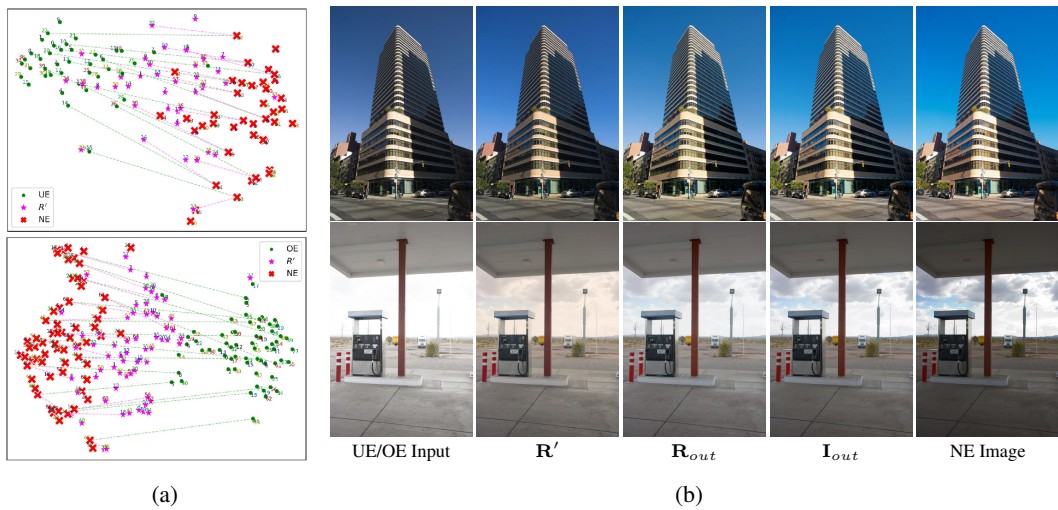

| UE/OE Input | $\mathbf{R}'$ | $\mathbf{R}_{out}$ | $\mathbf{I}_{out}$ | NE Image |

(a)                (b)

Figure 1: (a) T-SNE [7] visualization of distributions of modulated reflectance ($\mathbf{R}'$), restored reflectance ($\mathbf{R}_{out}$) and the final output ($\mathbf{I}_{out}$) for Under-Exposed (UE) and Over-Exposed (OE) images. Compared to the input data, modulated reflectance ($\mathbf{R}'$) demonstrates closer approximation of Normal-Exposed (NE) images. Besides, compared to the restored reflectance ($\mathbf{R}_{out}$), our final output ($\mathbf{I}_{out}$) are better aligned with NE data. (b) Visual result of $\mathbf{R}'$, $\mathbf{R}_{out}$ and $\mathbf{I}_{out}$ produced by our method. From column 2-4, we observe a noticeable improvement on color preservation and structure recovery, which demonstrates the importance of our introduced two-branch Retinex-based pipeline and the effectiveness of our proposed **ECMamba** network.

Retinex theory [22] has not yet been deeply integrated into deep learning models for multi-exposure correction. As deep learning models often struggle to distinguish between illumination information and the intrinsic reflectance properties of objects in images, simply adopting deep learning models to address such a difficult problem usually obtains sub-optimal results and the incorporation of Retinex theory offers a physically justified way to decompose the illumination and reflectance within deep learning models. Moreover, current state-of-the-art (SOTA) performance is achieved through introducing specific designs (exposure normalization [19] or exposure regularization term [21]) to existing networks. However, these methods present limited generalization, it is essential to develop stronger foundational model with good generalizable ability. Additionally, many methods face a trade-off between performance and efficiency, particularly those based on transformers.

To address these issues, we introduce a novel two-branch **E**xposure **C**orrection network (**ECMamba**) based on standard **Mamba** architecture, which has demonstrated impressive sequence modeling ability with high efficiency [14]. Specifically, we derive the Retinex theory and develop a Retinex estimator to transform the input into two intermediary spaces, each approximating the target reflectance and illumination map, respectively. As shown in Fig. 1a, compared to the input distribution, the generated intermediary space ($\mathbf{R}'$) shows closer approximation of target distribution, thus enabling subsequent network to execute the fine-grained restoration. Moreover, the visually compelling results in Fig. 1b column 4 demonstrate that our proposed two-branch framework offers more precise estimation and improved performance than simply optimizing the reflectance. Furthermore, we develop a novel **2D S**elective **S**tate-space layer guided by **Retinex** information (**Retinex-SS2D**) as the core operator of our ECMamba. Different from other scan strategies (*i.e.,* cross-scan mechanism [27]) which considers the scanning of 2D data to 1D sequence a "direction-sensitive" problem, we regard this issue as a "feature-sensitive" problem. Therefore, we first perform feature fusion and then introduce a **D**eformable **F**eature **A**ggregation (**DFA**) guided by Retinex information. Then based on the activation response map derived from DFA, we develop a **F**eature-**A**ware **2D S**elective **S**canning (**FA-SS2D**) mechanism to flatten the aggregated feature into 1D sequence, which is subsequently fed into the standard Selective State Space process (S6) to capture long-range dependencies.

The contributions of this work are summarized as follows:

⋄ We present a novel **dual-branch framework** that fully embeds **Retinex theory** for exposure correction and we provide detailed explanation for its significance.

⋄ By analyzing the operating mechanism of Selective State Space Model, we regard the scanning of vision data is a **"feature-sensitive"** issue and we propose an **efficient Retinex-SS2D layer** with Retinex-guided Feature-Aware 2D Selective Scanning Mechanism.

⋄ Extensive experiments and ablations demonstrate the **impressive performance** of our proposed method.

## 2   Related Works

**Learning based Multi-Exposure Correction**   Multi-exposure correction is a challenging task due to the opposite optimization flows of under-exposure and over-exposure correction. MSEC [1] introduces a Laplacian pyramid architecture to restore lightness and structures. Later, several normalization and regularization methods [4, 21, 19, 20] are proposed for exposure correction. For example, the exposure normalization [19] is proposed for exposure compensation, ECLNet [21] introduces exposure-consistency representations with bilateral activation mechanism, and FECNet [20] opts to correct illumination in the frequency domain. Different from previous methods, we aim to develop a Retinex-based network, where Retinex guidance is utilized to modulating the optimization flows of under-exposure and over-exposure correction.

**State Space Model (SSMs)**   Due to its impressive modeling capability for long-range dependencies and its promising efficiency, State Space Models (SSMs) and recent proposed Structured State-Space Sequence model (S4) [15] has attracted great interests among researchers. Based on S4, several models and strategies are introduced to improve the efficiency and boost the capability, among which Mamba [14] introduces an input-dependent SSM with selective mechanism and achieves superior performance than Transformers for natural language processing. Moreover, some pioneering works have applied Mamba on vision task such as image segmentation [27], classification [47] and even restoration [17, 29]. We are the first to address exposure correction problem based on Mamba, and we innovatively introduce an efficient feature-aware scanning strategy in this work.

## 3   Preliminaries

**State Space Model (S4)**   State Space Model (S4) is introduced by combining recurrent neural networks (RNNs), convolutional neural networks (CNNs), and classical state space models. Specifically, for a sequence of $L$ length $\mathbf{x} \in \mathbb{R}^L$, the input at any time step $x(t) \in \mathbb{R}$ can be mapped to an output $y(t) \in \mathbb{R}$ through the following state space modeling:

$$
\begin{aligned}
h'(t) &= \mathbf{A}h(t) + \mathbf{B}x(t), \\
y(t) &= \mathbf{C}h(t),
\end{aligned}
\tag{1}
$$

where $\mathbf{h}(t) \in \mathbb{R}^{N \times 1}$ represents latent state and $N$ denotes the dimension scaling ratio in latent state. $\mathbf{A} \in \mathbb{R}^{N \times N}$, $\mathbf{B} \in \mathbb{R}^{N \times 1}$, and $\mathbf{C} \in \mathbb{R}^{1 \times N}$ are state transition matrix, control matrix, and output matrix, respectively. Mathematically, the differential equation in Eq. 1 has an equivalent integral equation and it can be solved using numerical computation. In order to integrating state space modeling into deep learning models, the discretization is required to convert continues-time model to discrete-time system by introducing the timescale $\Delta \in \mathbb{R}$. Specifically, the zero-order hold (ZOH) rule, which is commonly used in SSM-based deep learning algorithms, is applied to transform continuous parameters $\mathbf{A}, \mathbf{B}$ in Eq. 1 to discrete matrix $\bar{\mathbf{A}}, \bar{\mathbf{B}}$ as follows:

$$
\begin{aligned}
\bar{\mathbf{A}} &= \exp(\Delta \mathbf{A}), \quad \bar{\mathbf{B}} = (\Delta \mathbf{A})^{-1}(\exp(\mathbf{A}) - \mathbf{I}) \cdot \Delta \mathbf{B}, \\
h(t) &= \bar{\mathbf{A}}h(t-1) + \bar{\mathbf{B}}x(t), \quad y(t) = \mathbf{C}h(t),
\end{aligned}
\tag{2}
$$

where $\bar{\mathbf{A}} \in \mathbb{R}^{N \times N}$, $\bar{\mathbf{B}} \in \mathbb{R}^{N \times 1}$. Moreover, given a sequence with dimension $D$ and length $L$, the SSM is applied independently to each dimension and $B, C, \Delta$ are extended with an extra dimension $D$. The overall computation complexity is $O(LDN)$, which is linear to the sequence length.

**Selective State-Space Model (S6)**   Selective State Space Model is introduced in Mamba with a selective mechanism so that the parameters in SSM can dynamically select necessary information from the context. Specifically, $\bar{\mathbf{B}}, \mathbf{C}, \Delta$ are designed as input-dependent parameters by utilizing linear functions and broadcast operation. This selective mechanism can help Mamba effectively filtering

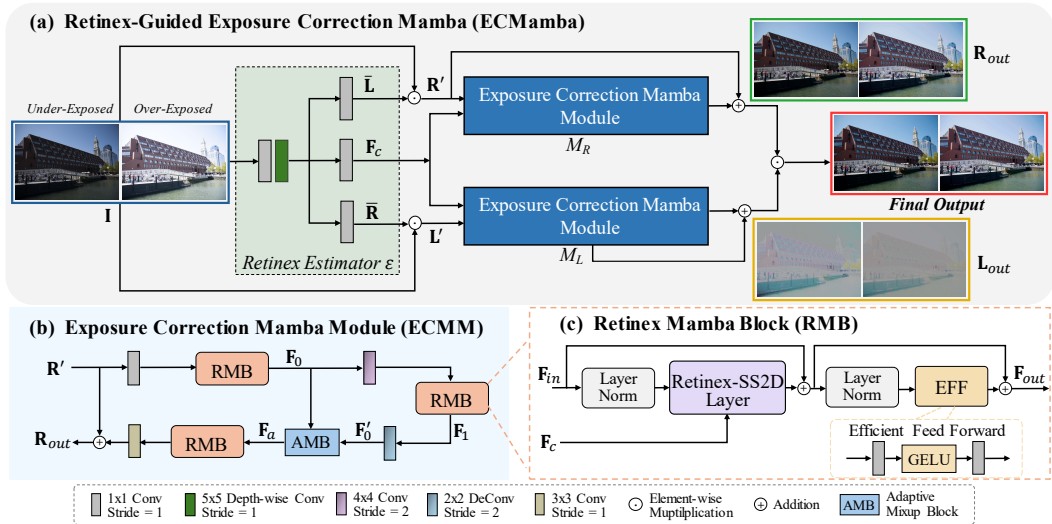

Figure 2: The overall architecture of our proposed Retinex-based framework for exposure correction, which includes a Retinex estimator $\mathcal{E}$ and primary restoration network $\mathcal{M}_R$ and $\mathcal{M}_L$.

out irrelevant noise and focusing on important tokens, thereby achieving outstanding performances in multiple language and vision tasks.

# 4 Methodology

The overall framework of our method is shown as Fig. 2, which demonstrates that our proposed exposure correction network is designed based on Mamba and Retinex theory. In this section, we first introduce our formulated Retinex-based Exposure Correction Framework (Sec. 4.1), then we propose to utilize **E**xposure **C**orrection **M**amba **M**odule (**ECMM**) to achieve precise restoration for the reflectance and illumination map (Sec. 4.2). More importantly, to enhance the efficiency and effectiveness, we introduce a new **2D S**elective **S**tate-space layer with an innovative scanning mechanism guided by **Retinex** information (**Retinex-SS2D**) in Sec. 4.3 and Sec. 4.4.

## 4.1 Retinex-Guided Exposure Correction Framework

The Retinex theory can be expressed as $\mathbf{I}_{GT} = \mathbf{R}_{GT} \odot \mathbf{L}_{GT}$, where $\odot$ denotes Hadamard product, $\mathbf{I}_{GT}$ is an ideal image without degradation, $\mathbf{R}_{GT}$ and $\mathbf{L}_{GT}$ represents the reflectance image and illumination map, respectively. However, a low-quality image $\mathbf{I}_{LQ}$ captured under non-ideal illumination conditions (under-exposure or over-exposure scenes) inevitably suffers from severe noise, color distortion, and constrained contrast. Therefore, we introduce a perturbation to $\mathbf{R}_{GT}$ and $\mathbf{L}_{GT}$ respectively ($\hat{\mathbf{R}}$ and $\hat{\mathbf{L}}$) to model these degraded images as:

$$\mathbf{I}_{LQ} = (\mathbf{R}_{GT} + \hat{\mathbf{R}}) \odot (\mathbf{L}_{GT} + \hat{\mathbf{L}}) = \mathbf{R}_{GT} \odot \mathbf{L}_{GT} + \mathbf{R}_{GT} \odot \hat{\mathbf{L}} + \hat{\mathbf{R}} \odot \mathbf{L}_{GT} + \hat{\mathbf{R}} \odot \hat{\mathbf{L}}. \quad (3)$$

Some existing Retinex-based methods [6, 13, 18, 33] regard the reflectance component $\mathbf{R}_{GT}$ as the final enhanced result, thus they ignore the last three terms in Eq. 3 and focus on modeling the mapping: $\mathbf{R}_{GT} = \mathcal{F}(\mathbf{I}_{LQ}) \odot (\mathbf{I}_{LQ})$ using network $\mathcal{F}$. However, these models can only achieve sub-optimal performance due to the difficulty of acquiring accurate mapping, especially for multiple exposure correction task, where multiple Under-Exposed (UE) and Over-Exposed (OE) inputs correspond to one Normal-Exposed (NE) image. Therefore, we choose to restore the reflectance and illumination component simultaneously in order to obtain satisfactory outputs. Specifically, we element-wisely multiply the both sides of Eq. 3 by $\bar{\mathbf{L}}$ and $\bar{\mathbf{R}}$ respectively as:

$$
\begin{aligned}
\mathbf{I}_{LQ} \odot \bar{\mathbf{L}} = \mathbf{R}' = \mathbf{R}_{GT} + \mathbf{R}_{GT} \odot \hat{\mathbf{L}} \odot \bar{\mathbf{L}} + \hat{\mathbf{R}} + \hat{\mathbf{R}} \odot \hat{\mathbf{L}} \odot \bar{\mathbf{L}}, \\
\mathbf{I}_{LQ} \odot \bar{\mathbf{R}} = \mathbf{L}' = \mathbf{L}_{GT} + \hat{\mathbf{R}} \odot \mathbf{L}_{GT} \odot \bar{\mathbf{R}} + \hat{\mathbf{L}} + \hat{\mathbf{R}} \odot \hat{\mathbf{L}} \odot \bar{\mathbf{R}},
\end{aligned}
\quad (4)
$$

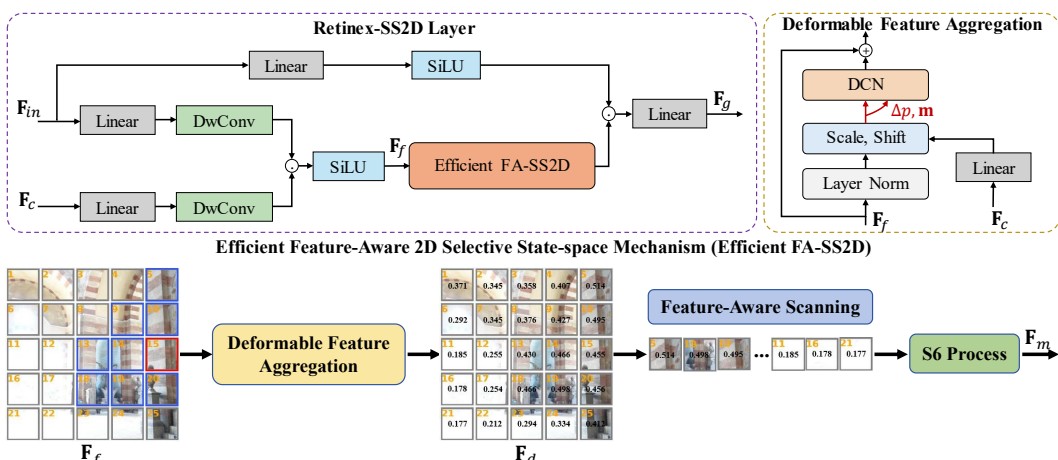

Figure 3: The details of our proposed Retinex-SS2D layer. We firstly fuse the input feature $\mathbf{F}_{in}$ and the Retinex guidance $\mathbf{F}_{in}$. Then we propose an innovative Feature-Aware 2D Selective State-spce Mechanism, which utilizes Deformable Convolution (DCN) for feature aggregation. Then we propose the feature-aware scanning strategy based on the activation response map derived from DCN. Compared to other 2D scanning methods, our approach generates a sequence ordered by feature importance, thereby maximizing the robust sequence modeling capabilities of Mamba.

where $\bar{\mathbf{L}}$ and $\bar{\mathbf{R}}$ are the matrix such that $\bar{\mathbf{L}} \odot \mathbf{L}_{GT} = \mathbf{1}$ and $\bar{\mathbf{R}} \odot \mathbf{R}_{GT} = \mathbf{1}$, and we assume we can approximate $\bar{\mathbf{L}}$ and $\bar{\mathbf{R}}$ via Retinex estimator $\mathcal{E}$. $\mathbf{R}_{GT} \odot \hat{\mathbf{L}} \odot \bar{\mathbf{L}} + \hat{\mathbf{R}} + \hat{\mathbf{R}} \odot \hat{\mathbf{L}} \odot \bar{\mathbf{L}}$ and $\hat{\mathbf{R}} \odot \mathbf{L}_{GT} \odot \bar{\mathbf{R}} + \hat{\mathbf{L}} + \hat{\mathbf{R}} \odot \hat{\mathbf{L}} \odot \bar{\mathbf{R}}$ indicate the remaining degradation in $\mathbf{R}'$ and $\mathbf{L}'$. Therefore, the well-exposed result can be retrieved using deep-learning networks by:

$$(\bar{\mathbf{R}}, \bar{\mathbf{L}}, \mathbf{F}_c) = \mathcal{E}(\mathbf{I}_{LQ}), \qquad \mathbf{R}' = \mathbf{I}_{LQ} \odot \bar{\mathbf{L}}, \qquad \mathbf{L}' = \mathbf{I}_{LQ} \odot \bar{\mathbf{R}},$$
$$\mathbf{R}_{out} = \mathbf{R}' + \mathcal{M}_R(\mathbf{R}'; \mathbf{F}_c), \quad \mathbf{L}_{out} = \mathbf{L}' + \mathcal{M}_L(\mathbf{L}'; \mathbf{F}_c), \quad \mathbf{I}_{out} = \mathbf{R}_{out} \odot \mathbf{L}_{out}, \tag{5}$$

where $\mathcal{M}_R$ and $\mathcal{M}_L$ are networks utilized to predict the minus degradation in $\mathbf{R}'$ and $\mathbf{L}'$, and $\mathbf{F}_c$ serves as a Retinex guidance information derived from the $\mathbf{I}_{LQ}$.

As shown in Fig. 2, the Retinex estimator $\mathcal{E}$ takes $\mathbf{I}_{LQ}$ and its mean matrix along the channel dimension (which is omitted for clarity in Fig. 2) as inputs. $\mathcal{E}$ firstly utilizes a $1 \times 1$ convolution and a depth-wise convolution with $5 \times 5$ kernel to extract features. Then, $\bar{\mathbf{L}}$, $\bar{\mathbf{R}}$ and $\mathbf{F}_c$ are generated by one $1 \times 1$ convolution, respectively. More importantly, $\mathbf{R}'$ and $\mathbf{L}'$ are fed into $\mathcal{M}_R$ and $\mathcal{M}_L$ for further restoration. In addition to optimizing $\mathbf{I}_{out}$ to approximate $\mathbf{I}_{GT}$, our training objective incorporates a constraint on $\bar{\mathbf{L}}$ and $\bar{\mathbf{R}}$, as discussed in Sec 4.5.

**Discussion** **(i)** Many Retinex-based methods [37] aim to learn the mapping from the input to the reflectance image and illumination map, then obtain the final result using Hadamard product operation. However, this strategy is not suitable for multi-exposure correction task. Fig. 1a illustrates the complicated and distant distribution patterns of OE and UE images relative to their normally exposed equivalents. Such complex distributions challenge the establishment of accurate mappings from inputs. However, by carefully analyzing Retinex theory, we construct an intermediary space that significantly reduces the distance to our optimization objectives and facilitates the subsequent fine-tuning restoration process, as shown in Fig.1b. **(ii)** Some methods [6, 13, 33, 18] treat $\mathbf{R}_{GT}$ as the final enhanced result, which deviates from the original explanation of Retinex theory and leads to limited performance. Therefore, we adopt a two-branch framework to reconstruct the reflectance and illumination map using distinct deep learning networks. The significance of our framework is discussed in the ablation study (Sec. 5.3).

## 4.2 Exposure Correction Mamba Module (ECMM)

Together with our proposed Retinex-guided exposure correction framework, we also develop innovative networks that serves as $\mathcal{M}_R$ and $\mathcal{M}_L$ in Eq. 5 to estimate the remaining corruption in $\mathbf{R}'$ and $\mathbf{L}'$. In order to develop an effective and efficient module that is capable to achieve high performance and is friendly to resource-limited devices, we propose an novel Retinex-guided **E**xposure **C**orrection

**Mamba M**odule (**ECMM**) which succeeds the powerful modeling capability of Mamba. Notably, $\mathcal{M}_R$ and $\mathcal{M}_L$ share similar structure and we discuss the details of $\mathcal{M}_R$ in this section.

As illustrated in Fig. 2, our ECMM adopts a two-scale U-Net architecture. For the encoding process, the input $\mathbf{R}'$ is firstly processed by a $conv\ 3 \times 3$ and one **R**etinex**M**amba **B**lock (**RMB**) to generate the initial feature $\mathbf{F}_0$. Then the downsampling operation is achieved by one $4 \times 4$ convolution with stride 2, and the down-sampled feature is fed into another RMB to obtain the middle feature $\mathbf{F}_1$. For the decoding stage, $\mathbf{F}_1$ is firstly up-scaled to $\mathbf{F}'_0$ by a $2 \times 2$ $deconv$ with stride 2. To alleviate the information loss caused by the down-sampling process, we introduce an adaptive mix-up feature fusion [45] to transfer the encoding information to the decoder stage as:

$$\mathbf{F}_a = \sigma(\theta)\mathbf{F}_0 + (1 - \sigma(\theta))\mathbf{F}'_0, \tag{6}$$

where $\theta$ represents a learnable coefficient, and $\sigma$ denotes the sigmoid function. Then the fused feature $\mathbf{F}_a$ is fed into the RMB and the convolution layer sequentially and the restored reflectance $\mathbf{R}_{out}$ is obtained by a residual addition accorrding to Eq. 5.

### 4.3 RetinexMamba Block (RMB)

As the core operator to extract and aggregate features in ECMM, our RMB block adopts a similar architecture with Transformer block. However, the significant computational demands of self-attention and cross-attention mechanisms obviously compromise the efficiency of Transformer-based methods, precluding their application in real-time or resource-constrained environments. To this end, we remove the attention process and introduce an innovative **R**etinex-guided **2D S**elective **S**tate-space (**Retinex-SS2D**) layer to capture long-range dependencies and facilitate dynamic feature aggregation. Therefore, the feature flow of our RMB can be described as:

$$\mathbf{F}'_{out} = \mathbf{F}_{in} + \text{Retinex-SS2D}(\text{LN}(\mathbf{F}_{in}), \mathbf{F}_c), \quad \mathbf{F}_{out} = \mathbf{F}'_{out} + \text{EFF}(\text{LN}(\mathbf{F}'_{out})), \tag{7}$$

where LN denotes the LayerNorm, $\mathbf{F}_{in}$ and $\mathbf{F}_{in}$ represent the input and output feature of RMB. $\mathbf{F}_c$ is the Retinex guidance information extract by the Retinex estimator $\mathcal{E}$. Moreover, inspired by ConvNext [28, 8], we remove the gating mechanism and the depth-wise convolution to develop an **E**fficient **F**eed **F**orward (**EFF**) layer that follows the $conv\ 1 \times 1 \rightarrow GELU \rightarrow conv\ 1 \times 1$ flow, which operates similarly to MLPs in Transformers, while requiring fewer parameters.

### 4.4 Retinex-SS2D Layer

The detailed illustration of Retinex-SS2D layer is shown as Fig. 3. We first conduct feature fusion for input feature $\mathbf{F}_{in}$ and Retinex guidance feature $\mathbf{F}_c$ by linear operation, depth-wise convolution, element-wisely multiplication, and SiLU operation. Subsequently, the fused feature $\mathbf{F}_f$ is fed into our proposed **F**eature-**A**ware **2D S**elective **S**tate-space (**FA-SS2D**) mechanism to capture dynamic long-range dependencies and achieve adaptive spatial aggregation. Besides, a gating signal $\mathbf{G}_s$ and a linear operation is utilized to obtain the aggregated feature $\mathbf{F}_g$.

**Feature-Aware 2D Selective State-space Mechanism** The standard Selective State-space Model (S6) achieves outstanding performance on sequence modeling, especially for NLP task that involves temporal sequence. However, significant challenges arise when applying S6 to 2D image. To better modeling the spatial information in 2D images, several interesting works propose multiple scan strategies to unfold image patches into 1D sequences. For example, [27] introduces cross-scan strategy that generate four sequences along four distinct traversal paths, and each sequence is processed by a separate S6 operation. However, such strategy incredibly increase the computational demands, which contradicts the inherently high efficiency and low computational requirements of S6. Furthermore, these techniques only involve simple scanning of images across different directions, which results in a substantial separation of local textures and global structures in some sequences. This separation, to some extent, impairs the S6 framework's modeling ability for images.

The deficiencies in the existing scanning approach drive us to reassess how S6 can be more effectively utilized for 2D images. As described in Eq. 2, for each token in a sequence, the output $y(t)$ depends on its input $x(t)$ and previous inputs $\{x(1), x(2), \cdots, x(t-1)\}$. This mechanism requires the 1D sequences transformed from 2D image to meet the following two criteria to ensure excellent performance: **(1)** The sequence should prioritize the most critical feature regions at the beginning, while relegating less significant information to the end. **(2)** Spatially adjacent features should be

| Methods | ME Dataset [1] | | | | | | SICE Dataset [5] | | | | | |
|---|---|---|---|---|---|---|---|---|---|---|---|---|
| | Under-exposed | | Over-exposed | | Average | | Under-exposed | | Over-exposed | | Average | |
| | PSNR↑ | SSIM↑ | PSNR↑ | SSIM↑ | PSNR↑ | SSIM↑ | PSNR↑ | SSIM↑ | PSNR↑ | SSIM↑ | PSNR↑ | SSIM↑ |
| ZeroDCE [16] CVPR'20 | 14.55 | 0.589 | 10.40 | 0.5142 | 12.06 | 0.544 | 16.92 | 0.633 | 7.11 | 0.429 | 12.02 | 0.531 |
| RUAS [24] CVPR'21 | 13.43 | 0.681 | 6.39 | 0.466 | 9.20 | 0.552 | 16.63 | 0.559 | 4.54 | 0.320 | 10.59 | 0.439 |
| URetinexNet [37] CVPR'22 | 13.85 | 0.737 | 9.81 | 0.673 | 11.42 | 0.699 | 17.39 | 0.645 | 7.40 | 0.454 | 12.40 | 0.550 |
| KinD [44] MM'19 | 15.51 | 0.761 | 11.66 | 0.730 | 13.20 | 0.742 | 13.43 | 0.484 | 7.85 | 0.478 | 10.64 | 0.481 |
| LLFlow* [34] AAAI'22 | 22.35 | 0.858 | 22.46 | 0.863 | 22.42 | 0.861 | 21.45 | 0.679 | 20.29 | 0.671 | 20.87 | 0.675 |
| LLFLow-SKF* [38] CVPR'23 | 22.58 | 0.859 | 22.72 | 0.865 | 22.66 | 0.863 | 21.61 | 0.671 | 20.55 | 0.695 | 21.08 | 0.683 |
| DRBN [40] CVPR'20 | 19.74 | 0.829 | 19.37 | 0.832 | 19.52 | 0.831 | 17.96 | 0.677 | 17.33 | 0.683 | 17.65 | 0.680 |
| DRBN+ERL [21] CVPR'23 | 19.91 | 0.831 | 19.60 | 0.838 | 19.73 | 0.836 | 18.09 | 0.674 | 17.93 | 0.687 | 18.01 | 0.680 |
| FECNet [20] ECCV'22 | 22.96 | 0.860 | 23.22 | 0.875 | 23.12 | 0.869 | 22.01 | 0.674 | 19.91 | 0.696 | 20.96 | 0.685 |
| FECNet+ERL [21] CVPR'23 | 23.10 | 0.864 | 23.18 | 0.876 | 23.15 | 0.871 | 22.35 | 0.667 | 20.10 | 0.689 | 21.22 | 0.678 |
| Retiformer* [6] ICCV'23 | 22.77 | 0.862 | 22.24 | 0.860 | 22.45 | 0.861 | 22.15 | 0.665 | 20.21 | 0.669 | 21.18 | 0.667 |
| LACT [4] ICCV'23 | 23.49 | 0.862 | 23.68 | 0.872 | 23.57 | 0.869 | - | - | - | - | - | - |
| Ours | 23.64 | 0.875 | 23.84 | 0.882 | 23.76 | 0.879 | 22.87 | 0.745 | 21.23 | 0.727 | 22.05 | 0.736 |

Table 1: Quantitative comparisons of different methods on multi-exposure correction datasets. The best and second-best results are highlighted in **bold** and underlined, respectively. "↑" means the larger, the better. Note that we obtain these results either from the original papers, or by running the officially released pre-trained models. "*" means that original papers don't report corresponding performance and we train their models using their officially released code.

sequenced closely to avoid significant gaps in the sequence. However, existing 2D scanning strategies fail to meet these two requirements, motivating us to propose new solutions to address this gap.

Based on these observations, we introduce an efficient Feature-Aware 2D Selective State-space (FA-SS2D) mechanism, as shown in Fig. 3. Firstly, we develop a deformable feature aggregation operation modulated by Retinex information. Specifically, Deformable Convolution (DCN) [48, 9] is adopted to capture dynamic long range dependencies of the fused feature $\mathbf{F}_f$. For example, when DCN is applied to the token delineated by the red frame in $\mathbf{F}_f$ of Fig. 3, its receptive field is an irregular kernel and the activated tokens are outlined in blue. More importantly, when the red frame is sliding across the feature map, the irregular kernel varies and we record which tokens are activated. After this process, we obtain the total activation number and calculate the average activation frequency for each token. Therefore, we can obtain an activation response map shown in $\mathbf{F}_d$ of Fig. 3, where tokens with higher activation frequency represent important features. Specifically, relatively brighter areas in under-exposed images or relatively normally exposed areas in over-exposed images contain important features and exhibit a large activation response. Based on the obtained activation response map, we propose a feature-aware scanning strategy. Different from "direction-sensitive" scanning method [27], our feature-aware strategy ranks tokens by their activation frequency and place tokens with higher frequencies at the start of the sequence. Therefore, our generated sequence effectively meets Mamba's requirements, thereby maximizing its modeling capabilities for vision data.

### 4.5 Loss Functions and Constraints

In this work, we adopt the one-stage strategy to train our proposed ECMamba network, which means that the $\mathcal{E}$, $\mathcal{M}_R$, and $\mathcal{M}_L$ are optimized simultaneously. Our ultimate training objective is to approximate $\mathbf{L}_{out}$ to $\mathbf{I}_{GT}$, and we also integrate several constraints on $\bar{\mathbf{L}}$, $\bar{\mathbf{R}}$, and $\mathbf{R}_{out}$ to achieve stable training. Therefore, our complete optimize strategy is shown as below:

$$\min_{\mathcal{E},\mathcal{M}_R,\mathcal{M}_L} \mathcal{L}(\mathbf{I}_{out}, \mathbf{I}_{GT}) + \lambda_L \cdot \mathcal{L}_1(\bar{\mathbf{L}} \odot \mathbf{L}_{out}, \mathbf{1}) + \lambda_R \cdot \mathcal{L}_1(\bar{\mathbf{R}} \odot \mathbf{R}_{out}, \mathbf{1}) + \lambda \cdot \mathcal{L}_1(\mathbf{R}_{out}, \mathbf{I}_{GT}), \quad (8)$$

where $\mathcal{L}_1(\bar{\mathbf{L}} \odot \mathbf{L}_{out}, \mathbf{1})$ and $\mathcal{L}_1(\bar{\mathbf{R}} \odot \mathbf{R}_{out}, \mathbf{1})$ are constraints applied on $\bar{\mathbf{L}}$ and $\bar{\mathbf{R}}$, and they essentially employ a self-supervised strategy to learn $\bar{\mathbf{L}}$ and $\bar{\mathbf{R}}$. In addition, considering this optimization is inherently an ill-posed problem, we adopt $\mathcal{L}_1(\mathbf{R}_{out}, \mathbf{I}_{GT})$ to guide the optimization towards the appropriate direction. Moreover, $\mathcal{M}_L \mathcal{L}(\mathbf{I}_{out}, \mathbf{I}_{GT})$ is the primary loss function in our training process and it can be calculated by:

$$\mathcal{L}(\mathbf{I}_{out}, \mathbf{I}_{GT}) = \mathcal{L}_1(\mathbf{I}_{out}, \mathbf{I}_{GT}) + \phi_{ssim} \cdot \mathcal{L}_{ssim}(\mathbf{I}_{out}, \mathbf{I}_{GT}) + \phi_{per} \cdot \mathcal{L}_{per}(\mathbf{I}_{out}, \mathbf{I}_{GT}), \quad (9)$$

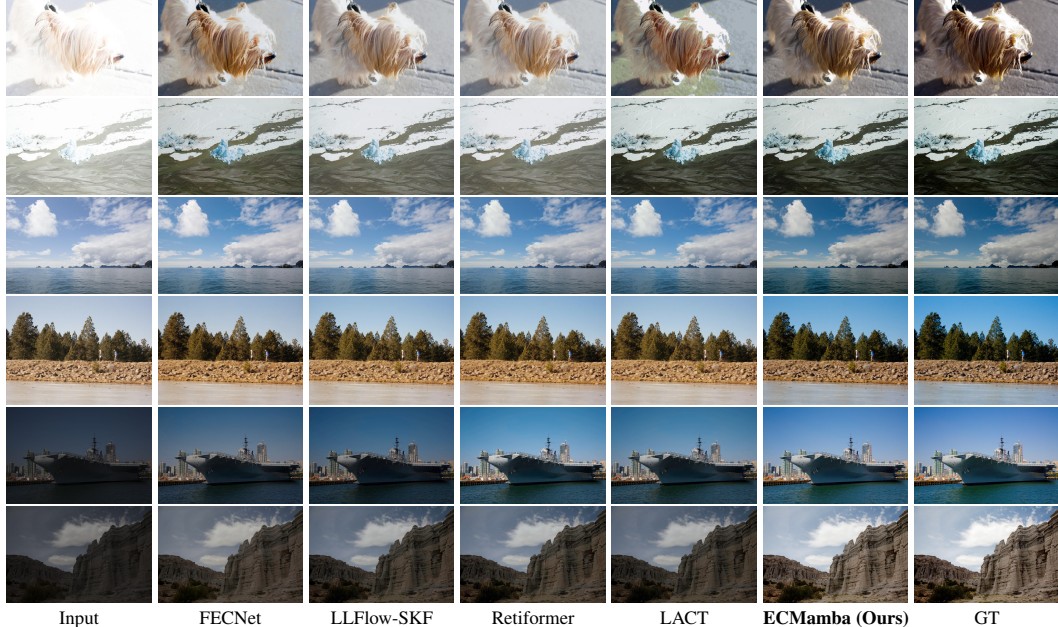

| Input | FECNet | LLFlow-SKF | Retiformer | LACT | **ECMamba (Ours)** | GT |

Figure 4: Visual comparison results on ME dataset. Compared to other exposure correction methods, our ECMamba excels in color preservation and structure recovery.

where $\mathcal{L}_{ssim}$ [46] denotes the structure similarity loss and $\mathcal{L}_{per}$ [43] represents the difference between features extracted by VGG19 [30]. The coefficients for corresponding loss functions are set as: $\phi_{ssim} = 0.2$, $\phi_{per} = 0.01$, $\lambda = 0.1$, $\lambda_R = 0.1$, and $\lambda_L = 0.1$ in this work.

## 5 Experiments

### 5.1 Experiment Settings

**Datasets**   To evaluate the performance of our method, we conduct experiments on five prevailing datasets for multi exposure correction and under-exposure correction: ME [1], SICE [5], LOLv1 [36], LOLv2-real [41], and LOLv2-synthetic [41] datasets. Specifically, each scene in ME dataset has five exposure levels, and we regard the images with the first two exposure level as under-exposed images and the test as over-exposed images. For SICE, following FECNet [20], we select the middle exposure subset as the ground truth, and define the second and the last second exposure subset as the under-exposed and over-exposed images, respectively. For LOLv1, LOLv2-real, LOLv2-synthetic datasets, we leverage their official training and testing data for model training and evaluation.

**Implementation Details**   We use the Adam optimizer with default parameters ($\beta_1 = 0.9$, $\beta_2 = 0.99$) to implement our model by PyTorch. The initial learning rate is set to $1 \times 10^{-4}$ and then it is steadily decreased to $1 \times 10^{-6}$ by the cosine annealing scheme, respectively. We utilize random flipping and rotation for data augmentation. Image pairs are cropped as $256 \times 256$ and the batch size is set to $4$. The total training iterations is set to 300K for ME dataset and 150K for other benchmarks, respectively. During the evaluation, we utilize Peak Signal-to-Noise Ratio (PSNR) and Structural Similarity Index Measure (SSIM [35]) for numeric evaluation.

### 5.2 Performances on Multi-Exposure and Under-Exposure Correction

**Quantitative Results**   We compare the performance of our ECMamba with current SOTA methods on multi-exposure correction datasets, and we report the quantitative results as Tab. 1. Notably, ECMamba significantly outperforms the current SOTA methods on both under-exposed and over-exposed images within ME dataset and SICE dataset. Specifically, our ECMamba excels in PSNR and SSIM, outperforming the second best method over 0.19 dB and 0.007 on ME dataset. Furthermore,

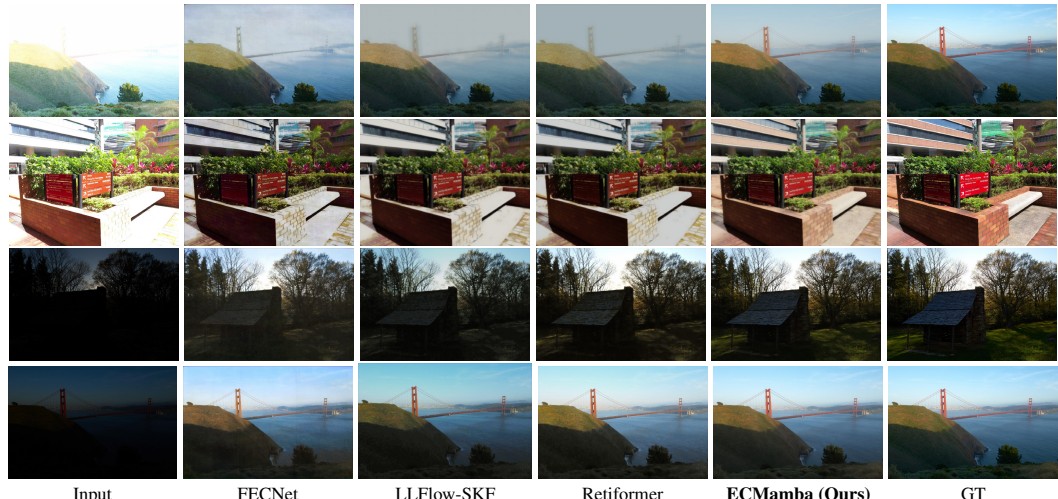

| Input | FECNet | LLFlow-SKF | Retiformer | **ECMamba (Ours)** | GT |

Figure 5: Visual comparisons between ECMamba and other methods on SICE dataset. Our proposed ECMamba achieves compelling visual performance both on over-exposed and under-exposed images.

| Methods | LOLv1 [36] PSNR↑ | SSIM↑ | LOLv2-real [41] PSNR↑ | SSIM↑ | LOLv2-synthetic [41] PSNR↑ | SSIM↑ | Param (M)↓ |
|---|---|---|---|---|---|---|---|
| Zero-DCE [16] CVPR'20 | 14.86 | 0.562 | 18.06 | 0.580 | - | - | 0.33 |
| RUAS [24] CVPR'21 | 18.23 | 0.720 | 18.37 | 0.723 | 16.55 | 0.652 | 0.003 |
| URetinex-Net [37] CVPR'22 | 21.33 | 0.835 | 21.16 | 0.840 | 24.14 | 0.928 | 1.32 |
| KinD [44] MM'19 | 20.86 | 0.790 | 14.74 | 0.641 | 13.29 | 0.578 | 8.02 |
| LLFlow [34] AAAI'22 | 25.19 | 0.870 | 26.53 | 0.892 | 26.08 | 0.940 | 37.68 |
| LLFlow-SKF [38] CVPR'23 | 26.80 | 0.879 | 28.19 | 0.905 | 28.86 | 0.953 | 39.91 |
| DRBN [40] CVPR'20 | 19.39 | 0.817 | 20.29 | 0.831 | 23.22 | 0.927 | 5.27 |
| DRBN+ERL [21] CVPR'23 | 19.84 | 0.830 | - | - | - | - | - |
| FECNet [20] ECCV'22 | 22.03 | 0.836 | 20.29 | 0.831 | 23.22 | 0.927 | 0.15 |
| FECNet+ERL [21] CVPR'23 | 21.08 | 0.829 | - | - | - | - | - |
| Retiformer [6] ICCV'23 | 25.16 | 0.845 | 22.80 | 0.840 | 25.67 | 0.930 | 1.61 |
| LACT* [4] ICCV'23 | 26.49 | 0.867 | 26.95 | 0.888 | 27.24 | 0.941 | 6.73 |
| **ECMamba (Ours)** | **27.69** | **0.885** | **29.24** | **0.908** | **29.94** | **0.959** | 1.75 |

Table 2: Quantitative comparisons of different methods for under-exposed correction. Notably, compared to SOTA methods, our ECMamba achieves enhanced performance on LOLv1 [36], LOLv2-real [41], and LOLv2-synthetic [41] datasets, demonstrating the effective of our proposed dual-branch Retinex-based framework and feature-aware SS2D layer.

compared to the second best performance, our improvement has increased to 0.83 dB and 0.051 on SICE dataset. Tab. 2 summarizes the quantitative comparisons between our method with current SOTA methods on on under-exposure correction. Specifically, our ECMamba outperforms the second best performance (LLFlow-SKF) by an average 1.10 dB increase on PNSR with only 4.4% parameters, revealing the impressive effectiveness and high efficiency of our proposed ECMamba. These numbers demonstrate the superior quality of our enhancement and prove the effectiveness of our proposed ECMamba and two-branch Retinex-based pipeline.

**Qualitative Comparisons**   We present the enhanced images of different methods in Fig. 4 (ME) and Fig. 5 (SICE). Our appealing and realistic enhancement results demonstrate our model can generate images with pleasant illumination, correct color retrieval, and enhanced texture details. For example, the rich structural details of cloud patterns (row 2) mountain surface (row 4) and in Fig. 4, the well preserved bridge and its edge contours (row 1) and the vivid presentation of words in the

| Methods | PSNR↑ | SSIM↑ | Param (M)↓ | Methods | PSNR↑ | SSIM↑ | Param (M)↓ |
|---|---|---|---|---|---|---|---|
| Removing $\mathcal{M}_L$ | 21.55 | 0.721 | 1.0 | ViT | 21.88 | 0.724 | 14.46 |
| Removing $\mathcal{M}_L^*$ | 21.63 | 0.723 | 1.93 | Retiformer | 21.35 | 0.702 | 1.6 |
| Removing $\mathcal{E}$ | 21.12 | 0.695 | 1.5 | Cross-Scan Mechanism | 21.69 | 0.716 | 2.1 |

Table 3: Ablation studies on SICE dataset and the average PSNR and SSIM are reported. *Left* is used to verify the effectiveness of the two-branch Retinex-based framework, *right* is to present the importance of our proposed ECMamba module and FA-SS2D strategy. [Key: $^*$: When $\mathcal{M}_L$ is removed, the hidden dimension of $\mathcal{M}_R$ is increased to ensure its parameter number is comparable to ECMamba; ViT/Retiformer: utilized to replace our ECMamba module; Cross-Scan Mechanism: adopted to replace our FA-SS2D strategy.]

display board (row 2) in Fig. 5. In contrast, previous methods struggle to preserve color fidelity and illumination harmonization.

## 5.3 Ablation Study

To verify the effectiveness of our proposed ECMamba, we conduct extensive ablation experiments and report the average performance on SICE dataset.

**The Contribution of Two-branch Retinex-based Framework**    We first remove the branch ($\mathcal{M}_L$), which is utilized for accurate restoration of the illumination map. Therefore, the remaining network aims to optimize $\mathbf{R}_{out}$ to the ground truth and the performance is reported in Tab. 3, which still presents competitive performance compared to the current SOTA in Tab. 1. However, compare to our complete ECMamba, only optimizing the relectance inevitably leads to sub-optimal performance. Furthermore, we also adopt a more complicated $\mathcal{M}_R$, whose parameters is comparable to the original two-branch framework. However, compared to our two-branch ECMamba, this network still demonstrate poor performance. Finally, similar to other Retinex-based methods [37], we then remove the Retinex estimator $\mathcal{E}$ and directly adopt the remaining network for exposure correction. However, as shown in Tab. 3, this adaptation largely decrease the performance of our ECMamba, indicating the importance of our analysis regarding the intermediary space ($\mathbf{R}'$ and $\mathbf{L}'$) in Sec. 4.1.

**The Importance of Our Proposed ECMamba Module**    First, we replace our ECMamba module with Vision Transformer (ViT) [11] and Retiformer [6] architecture in our two-branch framework. As presented in Tab. 3, our complete ECMamba module offers impressive performance better than ViT. More importantly, ECMamba's efficiency is comparable to Retiformer, which is a famous efficient under-exposure correction approach. Furthermore, to study the significance of our proposed Retinex-SS2D layer and FA-SS2D, we replace the Retinex-SS2D layer with a cross-scan mechanism proposed in VMamba [27]. The increased parameters and decreased performance demonstrate the superiority of our proposed Retinex-SS2D layer and FA-SS2D strategy.

## 6   Conclusions

We propose a new two-branch Retinex-based Mamba architecture for exposure correction. By carefully deriving Retinex theory, we propose an two-branch framework guided by Retinex information. To better balance the performance and efficiency, we introduce ECMamba as the primary restoration module with efficient Retinex-guided SS2D layer and Feature-aware scanning strategy. Extensive experiments demonstrate that our ECMamba significantly outperforms the current SOTA methods on both multi-exposure correction datasets and under-exposure correction datasets. We recognize that our work, while pioneering in certain aspects, also highlights avenues for future investigation. For example, similar to other methods, ECMamba struggles to deliver satisfactory results in scenarios involving extreme exposure cases (extremely dark or over-exposed environments) due to the extensive information loss inherent in degraded images. Essentially, directing recovery from severely degraded images is challenging, but recent advances in image restoration have utilized generative priors to infer the degraded details, and achieve favorable results. In the future, we plan to integrate Mamba with generative priors to effectively alleviate the performance drop on extreme exposure cases.

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
