# OpenReview forum: "ECMamba: Consolidating Selective State Space Model with Retinex Guidance for Efficient Multiple Exposure Correction"
_NeurIPS.cc/2024/Conference — NeurIPS 2024 poster_

### Official Review · Reviewer_zooA · 2024-07-08

**Soundness:** 3
**Presentation:** 3
**Contribution:** 3
**Rating:** 7
**Confidence:** 5

**Summary:**

The manuscript proposes a novel model that deeply embeds Retinex theory into the Mamba model. The proposed model consists of two modules: Retinex estimator and exposure correction. Comparative experiments on five datasets demonstrate the superiority of the proposed method, and subsequent ablation experiments demonstrated the rationality of the model construction.

**Strengths:**

Unlike previous work that used the R component of Retinex as the output, this model processed both the R and L components simultaneously, resulting in satisfactory results.

**Weaknesses:**

The experimental validation leaves much to be desired as I mention in the questions section.

**Questions:**

	In page 1, line 35, the authors stated that “Retinex theory has not been deeply integrated into…”, which could cause some negative effects? Please provide some explanations.
	The RMB in Figure 2 is mentioned in page 5, line 162, but the corresponding model diagram is missing. Please provide more explanations.
	In Figure 3, which arrows are marked with ∆p and m?
	In the ablation experiments, the author removed ML twice, what are the differences between those operations?
	In qualitative comparisons, the author do not list all the comparison experiment results, only list some results. For example, as can be seen in Table 1, the results of FECNet+ERL are better than FECNet, why not choose FECNet+ERL for displaying?

**Limitations:**

The authors do not point out the limitations of the work and do not offer further perspectives. I hope it will be improved.

---

> ### Author Rebuttal · Authors · 2024-08-03
>
> Thank you for your comprehensive and insightful review. We answer your questions in turn below:
>
> > **Q1:** In page 1, line 35, the authors stated that 'Retinex theory has not been deeply integrated into…', which could cause some negative effects? Please provide some explanations.
>
> Multi-exposure correction aims to correct improper illuminations in under/over-exposed images. It is quite challenging to simply adopt deep learning models to address this problem. This is because deep learning models often struggle to distinguish between illumination information and the intrinsic reflectance properties of objects in images. By incorporating Retinex theory to decompose the reflectance and illumination components, deep learning networks are more likely to produce high-quality enhancement results. Moreover, Retinex theory offers a physically justified way to interpret the decomposition of illumination and reflectance. Without Retinex theory, the inner working mechanisms of deep learning models may be more enigmatic and challenging to explain and understand.
>
> > **Q2:** The RMB in Figure 2 is mentioned in page 5, line 162, but the corresponding model diagram is missing. Please provide more explanations.
>
> In our manuscript, the diagram of RMB is provided in the orange dashed block in Figure 2(c). Its key component, the Retinex-SS2D layer (misspelled as Retinx-SS2D layer), is illustrated in Figure 3.
>
> > **Q3:** In Figure 3, which arrows are marked with $ ∆p $ and $ m $?
>
> The right arrow represents $ ∆p $ (offset) and $ m $ (modulation scalar),  which are obtained by applying a separate convolution over the output of the 'Scale, Shift' layer. Moreover, the $ ∆p $ and $ m $ are utilized to calculate the deformable features and the activation frequency map. We will make this clear in the revised version.
>
> > **Q4:** In the ablation experiments, the author removed $ \mathcal{M}_L $ twice, what are the differences between those operations?
>
> In our ablation study, we conduct two experiments related to the removal of $ \mathcal{M}_L $, denoted as 'Removing $ \mathcal{M}_L $' and 'Removing $ \mathcal{M}_L^* $', respectively.  'Removing $ \mathcal{M}_L^* $' means that while removing $ \mathcal{M}_L $, we also increase the hidden channel of the remaining network  $ \mathcal{M}_R $  to make its complexity similar to our full network. Through this ablation comparison under similiar complexity, we emphasize that the superior performance of our method is not attributed to the increased model parameters but to our proposed two-branch network, which optimizes both the reflectance and illumination map simultaneously.
>
> > **Q5:** In qualitative comparisons, the authors do not list all the comparison experiment results, only list some results. For example, as can be seen in Table 1, the results of FECNet+ERL are better than FECNet, why not choose FECNet+ERL for displaying?
>
> We report quantitative results for baseline models following their original papers. However, the paper that introduces ERL strategy doesn't release neither their visual results nor their implementation code, so we can't reproduce or access to their visual results. This is the reason why we didn’t report visual results for FECNet+ERL.
>
> > **Q6:** Limitations and future perspectives.
>
> We have briefly discussed the limitation of our method in the conclusion. Moreover, we add more discussions towards limitations and future perspectives in Q2 in the global response. We plan to incorporate this more detailed version into our revised manuscript.

---

### Official Review · Reviewer_EgGV · 2024-07-10

**Soundness:** 2
**Presentation:** 4
**Contribution:** 2
**Rating:** 5
**Confidence:** 4

**Summary:**

The authors propose a novel Mamba architecture for the exposure correction task based on Retinex theory. They design a separate Retinex estimator and two exposure correction modules to restore reflectance and illumination. In these modules, the authors improved efficiency in terms of time and resources by introducing the Mamba module. This model demonstrates excellent performance on the representative ME and SICE datasets.

**Strengths:**

* The mamba architecture is suitable for image processing because it can start processing even with partial input, similar to RNNs. From this perspective, the authors' attempt to introduce Mamba in the field of image restoration is technically very valid.
* Using the Selective State Space Model, the authors proposed the Retinex-SS2D layer, which enables Retinex-guided deformable feature aggregation.
* Unlike existing methods that aim to achieve limited generalization by introducing exposure normalization, the authors have developed a stronger model with good generalization capabilities.

**Weaknesses:**

* It is necessary to explain the design motivation of the Retina estimator. Unlike general networks, the authors start with a 1 by 1 convolution. It is necessary to explain the motivation behind designing such a network.
* A detailed explanation is needed on how \bar{L} and \bar{R} are constrained. R_gt and I_gt do not seem to be values that can be obtained directly.
* There is a lack of analysis and ablation studies on the network.
  * I am curious about the performance of the synthesized result when removing M_R and M_L, which are related to the terms overlooked in other papers. If I understand correctly, the performance should be similar to other works even without these two modules, especially since they are used in a residual form.
  * The comparison experiment between deformable convolution and standard convolution is missing.

**Questions:**

* The authors mention that the modulated reflectance (R′) demonstrates a closer approximation of Normal-Exposed (NE) images, but this is not well illustrated in Figure 1(a). If they could investigate and show the distribution of distances reduced via t-SNE, their claim would be more trustworthy.
* I am willing to increase the score if I have misunderstood something.

**Limitations:**

* This study is far from having a social impact. However, It is regrettable that the limitations and failure cases of the network's performance are not adequately addressed.

---

> ### Author Rebuttal · Authors · 2024-08-03
>
> Thank you for your review and comments. We provide our point-to-point response below:
>
> > **Q1:** The design motivation of the Retinex estimator
>
> The exposure correction task involves handling various exposure levels while also addressing complex issues such as color distortion and detail loss. Compared to employing an end-to-end approach to forcibly learn a unified mapping that simultaneously addresses these complex problems, the Retinex theory's decomposition of illumination and reflectance decouples the space of this mapping into two smaller subspaces. This facilitates better and easier optimization/learning, and enhances the interpretability of our model.
>
> > **Q2:** Why 1x1 convolution?
>
> Inspired by [R1], we first calculate the mean value for each pixel along the channel dimension and concat it to the original input within the Retinex estimator. Therefore, the 1x1 convolution is a good option to fuse the concatenation result across various channels. Also, compared to typical convolutions with large kernels, 1x1 convolution is more efficient. Besides, Retinex theory and later Retinex based methods [R2, R3] have underscored the independence of the reflectance and illumination map. To accurately decompose these two components, at the beginning of the network, it's better to operate independently on each pixel without considering the influence of its spatial neighbors.
>
> [R1] Deep Retinex Decomposition for Low-Light Enhancement. In BWVC 2018
>
> [R2] Lightness and Retinex Theory. In Journal of the Optical Society of America 1971.
>
> [R3] Diff-retinex: Rethinking low-light image enhancement with a generative diffusion model. In CVPR 2023.
>
> > **Q3:** Constraints on $\mathbf{\bar{L}}$ and $\mathbf{\bar{R}}$?
>
> Please see Q1 in our global response. We will add this part in our revised manuscript.
>
> > **Q4:** Performance when removing $\mathcal{M}_R$ and $\mathcal{M}_L$?
>
> The results after removing $ \mathcal{M}_R$ and $ \mathcal{M}_L $ are presented in Table C1. It is evident that the outcomes are significantly suboptimal, highlighting the indispensability of $ \mathcal{M}_R $ and $ \mathcal{M}_L $. Our motivation for designing $ \mathcal{M}_R $ and $ \mathcal{M}_L $ stems from the derivation provided in Equation 4 of our manuscript. This derivation indicates that even if the Intermediate Illumination Estimation Map $\mathbf{\bar{L}}$ can be accurately obtained—which could be approached by increasing the model complexity of the retinex estimator—the resultant $ \mathbf R^\prime = \mathbf R^{GT}+ \text{degradation terms}$. This implies that $\mathbf  R^{\prime} $ still contains non-negligible degradation terms, such as color distortion, as visually illustrated in Figure 1b. The same reasoning applies to $ \mathbf  L^{\prime} $.
>
> Therefore, $\mathcal{M}_R$ and $\mathcal{M}_L$ are primarily designed to eliminate these degradations. Without the degradation correction, performing an element-wise multiplication on the predicted reflectance and illumination, which contain these degradations, would amplify the deviations. Hence, $\mathcal{M}_R$ and $\mathcal{M}_L$ are essential as degradation correctors within the entire framework.
>
> **Table C1.** Performance of only $\mathcal{E}$ on SICE dataset.
> |Method|PSNR|SSIM|
> |-|-|-|
> |Only $\mathcal{E}$ |11.76|0.363|
> |ECMamba|22.05|0.736|
>
> > **Q5:** Performance when replacing deformable convolution with standard convolution
>
> Based on your suggestion, we conduct an ablation experiment by replacing the deformable convolution with standard convolution in our FA-SS2D mechanism, and the results are reported in Table C2. Evidently, the use of standard convolution yields inferior performance.
>
> Please note that deformable convolution is a crucial part of effectively applying the Mamba model to our vision task. Although Mamba has recently gained attention due to its efficient processing of long sequences in natural language processing, it was originally designed for one-dimensional data. This makes it less suitable for non-sequential two-dimensional images, which contain spatial information like textures and structures.
>
> To address this issue, we incorporated deformable convolution into the Mamba model. Unlike standard convolution, deformable convolution can dynamically learn appropriate receptive fields—either long or short—from the data. This ability to adaptively gather spatial information increases the likelihood of activating important image features. Based on this, we developed a strategy that prioritizes features according to their activation frequency, placing the most frequently activated features at the start of the sequence. This ensures each image patch is enriched with contextual knowledge, providing a new way to use Mamba in processing visual data.
>
> **Table C2.** Performance when replacing deformable convolution using standard convolution on SICE dataset.
> |Method|PSNR|SSIM|
> |-|-|-|
> |FA-SS2D with standard conv |21.45|0.709|
> |FA-SS2D with deformable conv (Ours)|22.05|0.736|
>
> > **Q6:** T-SNE visualization of distribution of reduced distance?
>
> We add new visualization to present the reduced distance of $\mathbf  R^{\prime} $ as Figure R4 in the attached PDF file in the global response. Please note that in order to make the visualization result more clear and readable, we randomly visualize part of images presented in Figure 1(a) of our manuscript. The green line represents the distance between the input and normal exposure features (NE), while the purple line indicates the distance between the modulated reflectance $\mathbf{R}^{\prime}$ and NE. We will add this new visualization in our revised version.
>
> > **Q7:** Limitations and failure case?
>
> We briefly discussed the limitation in the conclusion of our manuscript. We also provide more discussions for this part in Q2 in the global response.

---

> > ### Comment · Reviewer_EgGV · 2024-08-12
> > **Additional Comments**
> >
> > The authors have addressed my concerns through additional experiments. I will raise my score from 4 to 5. Thank you for your hard work during the rebuttal period.

---

> ### Comment · Area_Chair_3mTk · 2024-08-12
> **Would you please have a look at the rebuttal?**
>
> Dear Reviewer,
>
> Thanks a lot for contributing to NeurIPS2024.
>
> The authors have provided detailed responses to your review. Would you please have a look at them at your earliest convenience?
>
> Thanks again. AC

---

> ### Author Response · Authors · 2024-08-13
>
> Dear Reviewer EgGV,
>
> We are glad that our rebuttal has addressed your concerns. Thanks again for your valuable comments.
>
> Sincerely,
>
> Authors of Submission 3490

---

### Official Review · Reviewer_fpJd · 2024-07-10

**Soundness:** 4
**Presentation:** 3
**Contribution:** 4
**Rating:** 7
**Confidence:** 5

**Summary:**

This paper introduces a new pipeline called ECMamba for multiple exposure correction. Based on the analysis of Retinex theory, the authors develop a dual-branch framework, and each pathway is designed to restore the reflectance image and the illumination map, respectively. Besides, considering the powerful and efficient sequence modeling of the recently proposed Mamba, this paper also incorporates the Mamba architecture within each pathway and attempts to achieve effective and efficient exposure correction. To exploit Mamba to process image data, this paper develops an innovative "feature-aware" 2D scanning strategy based on deformable feature aggregation, which are quite different from other "direction-sensitive" scanning approaches. Finally, extensive experiments on multi-exposure and under-exposure datasets are conducted, and the reported results demonstrate the proposed method in this paper outperforms current SOTA approaches.

**Strengths:**

1) This paper is a retinex-based method. Authors carefully analyze the retinex component for corrupted images and then design a two-branch network based on their discussions, which makes the proposed method theoretically solid.

2) The T-SNE visualizations in Fig.1 and discussions in line 141-152 provide a detailed explanation for the design of the two-branch exposure correction network.

3) This paper deeply incorporates the retinex guidance into deep learning network, which is a promising way to learn a consistent transformation between under-/over-exposed inputs and normal-exposed images.

4) Different from simply regarding the scanning of images to 1D sequence as a "direction-sensitive" problem, this paper introduces an interesting concept — "feature-aware" scanning. This strategy is proposed by carefully analyzing the operation mechanism of Mamba, thus the"feature-aware" scanning strategy is theoretically sound and constructive.

5) Both quantitative and qualitative performance of the proposed method are impressive.

6) This paper is well written and easy to follow.

**Weaknesses:**

(1) This paper introduces an interesting two-branch framework based on retinex theory. Compared to a single-branch network, it is more difficult to train a two-branch network especially when the final result is obtained via Hadamard product. However, the constraint used in the training phase is less discussed in this paper.

(2) The proposed "feature-sensitive" scanning mechanism in Mamba is quite novelity. However, the difference in the activation frequency map between under-exposed images and over-exposed images is less explored.

(3) This paper adopts deformable convolution for feature aggregation and the generation of the the activation frequency map. However, how to get the activation frequency map is not very clear. For example in Fig.3, my understanding is that the 9 blue tokens are activated when a 3x3 deformable convolution is applied to the red token. Also, I think the numbers in F_d denote the average activtion frequency for each token when sliding the red token across the entire F_f. Am I correct?

Minor:
The reference [1] in line 194 is a typo, the reference should be Vmamba, which is cited as [18] in this paper.
"Deformable Feature Aggegration" in Fig.3 should be "Deformable Feature Aggregation".

**Questions:**

1) What constraint is adopted to train the proposed two-branch network? Is that possible that R_out and I_out are quite different form their corresponding ground truth, but their product I_out is very close to the well-exposed image I_GT?
2) What is the difference in the activation frequency map between under-exposed images and over-exposed images?
3) How to get the activation frequency map in Fig.3? Please explain in detail.

**Limitations:**

Authors have briefly discussed the limitations in Sec.6. There are no potential negative ethical and societal implications of this work.

---

> ### Author Rebuttal · Authors · 2024-08-04
>
> We sincerely thank you for the valuable comments on our paper. We will explain your concerns point by point.
>
> > **Q1:** What constraint is adopted to train the proposed two-branch network? Is it possible that $ \mathbf R_{out} $ and $ \mathbf I_{out} $ are quite different from their corresponding ground truth, but their product $ \mathbf I_{out} $ is very close to the well-exposed image $ \mathbf I_{GT} $?
>
> Thank you for your insightful reviews. It is true that training such two-branch network is more difficult, as there are no ground truth for the reflectance and illumination components for images. In our initial expriments, we only calculate the difference between $ \mathbf I_{out} $ and $ \mathbf I_{GT} $ as the loss function for optimization. However, we observe that $ \mathbf  R^{\prime} $, $ \mathbf  R_{out} $,  and $ \mathbf  L_{out} $ deviate their expected values evidently, though the final output $\mathbf  I_{out} $ is close to $ \mathbf I_{GT} $. We attribute this phenomenon to the fact that this optimization is an ill-posed problem. To tackle this issue, we attempt to add some constraints to the loss function, as shown in Q1 of our global response. With these constraints, we successfully achieve stable training for our ECMamba. We will add more discussion for constraints in our revised manuscript.
>
>
> > **Q2:** The proposed "feature-sensitive" scanning mechanism in Mamba is quite novel. However, what is the difference in the activation frequency map between under-exposed images and over-exposed images?
>
> We have visualized the activation frequency maps for under-exposed and over-exposed images in Figure D of the attached PDF. The difference in the activation frequency map is primarily caused by the different important feature regions in under-exposed and over-exposed images. Specifically, relatively brighter areas in under-exposed images or relatively normally exposed areas in over-exposed images contain important features and exhibit a large activation response in the corresponding activation frequency map.
>
>
> > **Q3** How to get the activation frequency map in Fig.3? Please explain in detail. My understanding is that the 9 blue tokens are activated when a 3x3 deformable convolution is applied to the red token. Also, I think the numbers in $ \mathbf F_d$ denote the average activation frequency for each token when sliding the red token across the entire F_f. Am I correct?
>
> Your understanding is correct. Different from typical convolution, the receptive field of 3x3 deformable convolution is an irregular kernel. For example, when applying deformable convolution for the red token in Figure 3, 9 blue tokens are activated and they form the irregular kernel. When the red token is sliding across the feature map, the irregular kernel varies and we will record which token is activated. After this process, we obtain the total activation number for each token and calculate the average activation frequency for each token, which is regarded as the activation frequency map.
>
> > **Q4** Incorrect reference and typo.
>
> Thanks for pointing this out. We will revise these mistakes and double check all over the manuscript.
>
> > **Q5** More discussions on Limitations of our paper.
>
> We appreciate your attention for our brief discussion on limitations in the conclusion of our manuscript. We also provide more discussions for this part in Q2 in the global response.

---

> > ### Comment · Reviewer_fpJd · 2024-08-12
> >
> > Thanks for your responses, which have well  addressed my concerns. Thus, I tend to keep my original score.

---

> ### Author Response · Authors · 2024-08-13
>
> Dear Reviewer fpJd,
>
> We appreciate your positive review and we are happy that our response well addressed your concerns.
>
> Thank you again for your valuable comments.
>
> Sincerely,
>
> Authors of Submission 3490

---

### Official Review · Reviewer_5Zdz · 2024-07-12

**Soundness:** 3
**Presentation:** 3
**Contribution:** 3
**Rating:** 5
**Confidence:** 5

**Summary:**

This paper introduces ECMamba, a novel framework that integrates Retinex theory and the Mamba framework to address the complex issue of exposure correction. ECMamba adapts the Retinex theory to suit the needs of exposure correction and develops a Retinex estimator to assess both reflectance and illumination maps. Subsequently, the framework employs Mamba to orchestrate the restoration process, termed ECMM. The ECMM module features a core operator, Retinex-SS2D, which employs a two-dimensional scanning strategy and deformable feature aggregation. The final output is generated by multiplying the enhanced reflectance map with the enhanced illumination map, yielding a superior result.

**Strengths:**

1. This paper represents the inaugural application of Retinex theory to the problem of exposure correction. Historically, Retinex theory was applied exclusively to low-light image enhancement, resulting in output images that are invariably brighter than their inputs. This study expands the theory's application, adapting it to scenarios where output images may be either brighter or darker than the inputs.

2. The paper is well-organized and communicates its concepts effectively.

**Weaknesses:**

1. The rationale for employing Retinex theory and the Mamba framework in multi-exposure correction is not adequately articulated. The author mentions that Retinex theory has not been deeply integrated into this field, yet this alone does not establish its superiority over alternative methods. A more robust justification for using Retinex theory, as well as the Mamba framework, is necessary to substantiate their relevance to solving this specific problem.

2. The initial two rows of Figure 4 are presumably intended to illustrate over-exposed scenarios. However, these images do not appear over-exposed as they lack saturated regions and display clear details and contrast. It would be advisable to replace them with images that more accurately represent over-exposure.

3. The inference speed of the proposed method is a critical aspect that is currently unaddressed in the paper.

**Questions:**

Shown in the "Weakness" part.

**Limitations:**

The paper does not discuss the limitations. No negative social impact is present in this work.

---

> ### Author Rebuttal · Authors · 2024-08-04
>
> Thank you for providing valuable feedback for our paper. We address your concerns in turn below. We hope our response can well address all your concerns.
>
> > **Q1:** The rationale for employing Retinex theory and the Mamba framework in multi-exposure correction is not adequately articulated. The author mentions that Retinex theory has not been deeply integrated into this field, yet this alone does not establish its superiority over alternative methods. A more robust justification for using Retinex theory, as well as the Mamba framework, is necessary to substantiate their relevance to solving this specific problem.
>
> Thank you for your insightful review. In addition to the fact that Retinex theory and the Mamba framework have not yet been applied to the multi-exposure correction task, the rationale for employing these methods can be justified from the following aspects:
>
> **Why is Retinex theory important?** Retinex theory, a physically validated theory, essentially studies the reflectance and illumination components of images, which is closely related to the task of correcting improper illuminations for over-/under-exposed images in this paper. Therefore, incorporating Retinex theory can help understand and solve multi-exposure correction problems, as well as enhance the interpretability of the algorithm.
>
> **Why is simply adopting Retinex theory not sufficient?** Simply adopting Retinex theory to decompose the reflectance and illumination component in a data-driven manner is quite challenging, as this problem is highly ill-posed. Therefore, through the derivation of Retinex theory, we introduce intermediate $\mathbf R^{\prime}$ and $\mathbf L^{\prime}$ and we enact models with powerful modeling capability (i.e., Mamba) as the primary network to remove the degradation in $\mathbf R^{\prime}$ and $\mathbf L^{\prime}$.
>
> **Motivation of utilizing Mamba.** One of the reasons is that recently proposed Mamba network demonstates impressive modeling for natural language and image data. But this is not the entire picture. Compared to the self-attention layer in transformers, whose computation complexity is quadratic to the sequence length, Mamba's recurrent working strategy and state-space mechanism require less computation that is linear to the sequence length. Please also refer to the Figure B in the PDF file attached in the global response. More importantly, we visualize the activation response maps in ViT for under-/over-exposed images in Figure B. These maps indicates that the activation response intensity from ill-posed areas to normally exposed regions is minimal. To optimize efficiency, we can exclude the computations of the response intensity from the ill-posed area to the relatively normally exposed region. Therefore, for muti-exposure correction task, the optimal working mechanism involves extracting critical information from the relatively normally exposed regions in under/over-exposed images to restore other areas. This working mechanism aligns perfectly with Mamba's recurrent functional principles. The remaining task is to prioritize these relatively normally exposed regions at the beginning of the sequence. Accordingly, we propose the feature-aware SS2D scanning mechanism based on deformable feature aggregation to achieve this objective. In this way, our Mamba network with feature-aware SS2D scanning mechanism is highly efficient and particularly well-suited for the specific task of exposure correction.
>
> **Why combine Mamba with Retinex theory?** In the ablation study, we have directly fed the input into Mamba networks, denoted as 'Removing $ \mathcal E $' in Table 3 in our manuscript, and we observe a 0.93 dB decrease in PSNR and 0.04 decline in SSIM compared to our complete ECMamba. This comparison justifies the importance of combining Mamba with Retinex theory.
>
>
> > **Q2:** The initial two rows of Figure 4 are presumably intended to illustrate over-exposed scenarios. However, these images do not appear over-exposed as they lack saturated regions and display clear details and contrast. It would be advisable to replace them with images that more accurately represent over-exposure.
>
> We appreciate your valuable suggestion. More appropriate visual comparisons are provided in Figure C of the rebuttal PDF file. We will replace previous images with these new visual results in our revised manuscript.
>
> > **Q3:** The inference speed of the proposed method is a critical aspect that is currently unaddressed in the paper.
>
> Thanks for pointing this out. In our manuscript, we have reported the model complexity of different models in Table 2. It is indeed important to provide additional metrics related to model efficiency, such as FLOPs and inference time. We have supplemented these metrics as shown in below Table A1. Thanks to Mamba's high efficiency and hardware-aware design, the efficiency of our method is comparable to or better than that of other models, except for FECNet.
>
> **Table A1.** Efficiency comparisons between our method and other models. Please note FLOPs is calculated with 256x256 inputs and the average inference time is calculated by testing with multiple 400x600 inputs on the same GPU device.
>
> |Metrics|KinD|DRBN|LLFlow|LLFlow-SKF|FECNet|Retiformer|LACT|ECMamba(Ours)|
> |--------- |-------|---------|----------|-------------------|----------|--------------|--------|-----------------------|
> |Parameters (M) |8.02|5.27|37.68|39.91|0.15|1.61|6.73|1.75|
> |FLOPs (G) |34.99|48.61|286.34|310.27|5.82|15.57|78.64|16.38|
> |Inference time (s)|1.136|0.228|0.399|0.474|0.099|0.189|0.295|0.164|
>
> > **Q4:** More discussions on limitations of our paper.
>
> We briefly addressed the limitations in the conclusion. Here, we provide a more thorough discussion of it in Q2 of our global response. We will update this in our revised manuscript.

---

> > ### Comment · Reviewer_5Zdz · 2024-08-13
> >
> > Thank you for your rebuttal. You have solved the majority of my concerns. Though I still have reservations about the motivation of utilizing Mamba and Retinex theory, I find most of the other explanations persuasive. Consequently, I am considering adjusting my rating to ``borderline accept``.

---

> ### Comment · Area_Chair_3mTk · 2024-08-12
> **Would you please have a look at the rebuttal?**
>
> Dear Reviewer,
>
> Thanks a lot for contributing to NeurIPS2024.
>
> The authors have provided detailed responses to your review. Would you please have a look at them at your earliest convenience?
>
> Thanks again.
> AC

---

> ### Author Response · Authors · 2024-08-13
>
> Dear Reviewer 5Zdz,
>
> Thanks again for your constructive comments. We are glad that our response resolved the majority of your concerns. We will add the explanations in the revised manuscript for more clarity.
>
> Sincerely,
>
> Authors of Submission 3490

---

### Author Rebuttal · Authors · 2024-08-04

We sincerely thank all reviewers (**R1** 5Zdz, **R2** fpJd, **R3** EgGV, and **R4** zooA) for their detailed reviews and constructive comments. The reviewers agree that:

**Novel or interesting approach:**
-  **R1:** "This study **expands the theory's application**, adapting it to scenarios..."
-  **R2:** "The proposed "feature-sensitive" scanning mechanism in Mamba is **quite novel**"

-  **R4:** "**Unlike previous work** that used the R component of Retinex as the output, this model..."

**Soild:**

-  **R2:** "Authors **carefully analyze** the retinex component..., which makes the proposed method **theoretically solid**. ..., thus the "feature-aware" scanning strategy is **theoretically sound and constructive**."

-  **R3:** "From this perspective, the authors' attempt to introduce Mamba in the field of image restoration is **technically very valid**."

**Well-written and Organized:**

-  **R1:**  "The paper is **well-organized** and communicates its concepts effectively."

-  **R2:**  "This paper is **well written** and easy to follow."

Here we first summarize the key clarifications in relation to some common review comments and concerns, and then address individually to each reviewer's concerns point by point.

**Q1: Detailed constraints applied on $ \mathbf {\bar{L}}$ and $ \mathbf {\bar{R}}$.**

In line 139-140, page 5 of our manuscript, we mentioned incorporating constraints on $ \mathbf {\bar{L}}$ and $ \mathbf {\bar{R}}$. It's challenging to directly optimize $ \mathbf {\bar{L}}$ and $ \mathbf {\bar{R}}$ due to the lack of their corresponding ground truths. Instead, we feed these two variables into subsequent networks to obtain the final enhanced image for optimization, and this strategy amounts to applying an indirect supervision to $ \mathbf {\bar{L}}$ and $ \mathbf {\bar{R}}$. Moreover, to achieve stable training, our complete optimize strategy is shown as below:

$$
\min_{ \mathcal E, \mathcal M_R, \mathcal M_L } \mathcal{L}(\mathbf I_{out}, \mathbf I_{GT}) + \lambda_L \cdot \mathcal L_1 (\mathbf{\bar{L}} \odot \mathbf L_{out}, \mathbf{1}) + \lambda_R \cdot \mathcal L_1 (\mathbf{\bar{R}} \odot \mathbf R_{out}, \mathbf{1}) + \lambda \cdot \mathcal L_1 (\mathbf R_{out}, \mathbf I_{GT}),
$$

where $\mathcal L_1 (\mathbf{\bar{L}} \odot \mathbf L_{out}, \mathbf{1})$ and $\mathcal L_1 (\mathbf{\bar{R}} \odot \mathbf R_{out}, \mathbf{1})$ are constraints applied on $ \mathbf {\bar{L}}$ and $ \mathbf {\bar{R}}$. These two constraints essentially employ a self-supervised strategy to learn $ \mathbf {\bar{L}}$ and $ \mathbf {\bar{R}}$. $\lambda_L$ and $\lambda_R$ are set to 0.1 in this work.

In addition, considering this optimization is inherently an ill-posed problem, we adopt $\mathcal L_1 (\mathbf  R_{out}, \mathbf I_{GT})$ to guide the optimization towards the appropriate direction and we set $\lambda$ to 0.1. Please note that $ \mathcal{L}(\mathbf I_{out}, \mathbf I_{GT}) $ is the primary loss function in our training process and it is calculated by the following equation:

$$
\mathcal{L}(\mathbf I_{out}, \mathbf I_{GT}) = \mathcal L_1(\mathbf I_{out}, \mathbf I_{GT}) + \phi_{ssim} \cdot \mathcal L_{ssim}(\mathbf I_{out}, \mathbf I_{GT}) + \phi_{per} \cdot \mathcal L_{per}(\mathbf I_{out}, \mathbf I_{GT}),
$$

where $ \mathcal L_{ssim} $ denotes the structure similarity loss and $ \mathcal L_{per} $ represents the difference between features extracted by VGG19. $ \phi_{ssim} = 0.2 $ and $ \phi_{per} = 0.01 $ are coefficients for corresponding loss functions.


**Q2: More discussions on limitations and future perspectives of our paper.**

 Here, we provide more detailed discussions about the limitation and future perspective of our paper:

**limitation:** While our method achieves impressive performance on various datasets, similar to other methods, ECMamba struggles to deliver satisfactory results in scenarios involving extreme exposure cases (extremely dark or over-exposed environments), as shown in Figure A in the attached PDF. Due to the extensive information loss inherent in such images, our method demonstrates limited capability of perfectly restoring them.

**Future perspective**: In extreme exposure scenarios, the image details are often degraded severely. Direct recovery is challenging, but recent advances in image restoration have utilized generative priors to infer the degraded details, and achieve favourable results. In the future, we plan to integrate Mamba with generative priors for LLIE to effectively alleviate the performance drop on extreme exposure cases.

---

### Decision · Program_Chairs · 2024-09-25

**Decision:**

Accept (poster)

**Comment:**

Following extensive interactions between the authors and reviewers, all reviewers have expressed a positive opinion on the paper. The AC believes the paper merits a clear acceptance. Although the authors have attempted to justify their use of Mamba and Retinex for this task, the rationale still lacks a solid foundation. The authors are encouraged to improve the paper by adhering to the commitments and clarifications made during the rebuttal and discussion